# A Survey of the Professional Characteristics and Views of Dog Trainers in Canada

**DOI:** 10.3390/ani15091255

**Published:** 2025-04-29

**Authors:** Camila Cavalli, Nicole Fenwick

**Affiliations:** 1Animal Welfare Program, Faculty of Land and Food Systems, The University of British Columbia, Vancouver, BC V6T 1Z4, Canada; 2British Columbia Society for the Prevention of Cruelty to Animals (BC SPCA), Vancouver, BC V5T 1R1, Canada; nfenwick@spca.bc.ca

**Keywords:** dog training methods, dog training credentials, regulation, dog welfare

## Abstract

Dog trainers are not regulated, officially certified, or required to follow guidelines in Canada. This raises concerns because trainers influence how guardians relate to their dogs and impact dog well-being. We carried out an online survey to learn more about dog trainers in Canada. We found that trainers held qualifications from a great variety of educational programs, which differed in their durations and the topics they covered. Reward-based training appears to be the most prevalent method in Canada, as most of the words or phrases trainers used to describe their training were related to this type of training, and most were unlikely to use aversive collars. Two-thirds of trainers were in favor of some regulation of dog training. The many types of courses, qualifications, and terms used, combined with the lack of professional oversight, make it challenging for guardians to choose a trainer who is knowledgeable and uses evidence-based training methods that promote dog welfare. The information in this study may help standardize training best practices and educational programs and inform advocacy efforts that promote humane dog training.

## 1. Introduction

Dog ownership is popular in Canada, with an estimated population of 7.9 million dogs [1]. Dog owners (i.e., dog guardians) often seek out the services of professional dog trainers for guidance with training and modification of unwanted behaviors. As a result, professional dog trainers have a great degree of influence over how guardians interact with their dogs and the welfare of these companion animals.

However, training is an unregulated profession in Canada, with no national or provincial registries of trainers, licensing, or standardization of practices and methods. There are also limited data about the dog training sector in Canada in terms of trainer demographics, educational background, services offered, training methods employed, and the views and opinions of dog trainers.

Both the lack of standardized training methods and the absence of professional regulation present challenges to dog guardians over which trainers to choose, and this can result in the use of training methods that harm dogs. The methods used to train a dog can be broadly classified as reward-based or aversive-based (e.g., [2]). Techniques based on the use of positive reinforcement and negative punishment are generally considered to be reward-based. Techniques that involve the use of stimuli that could be perceived as unpleasant by the dog are generally regarded as aversive-based and typically incorporate positive punishment and negative reinforcement [2,3]. Many trainers who incorporate aversive-based techniques also use rewards, and this blend of methods is colloquially referred to as “balanced training” (e.g., [4,5]).

Scientific findings consistently support the use of reward-based training due to its greater effectiveness and positive animal welfare outcomes (for reviews see [2,6,7]). In particular, the use of aversive-based methods has been associated with decreased animal welfare; increased aggression and fear (e.g., [8,9,10,11]); negative impacts on the human–animal bond, including decreased affiliative behaviors (e.g., [12,13,14]); and reduced training success (e.g., [13,15,16]).

Despite the scientific consensus, trainers differ greatly in their methods and underlying beliefs regarding dog behavior [17], and aversive-based techniques are still widely employed. Moreover, there is disagreement in both the literature and professional practices on which methods should be regarded as humane [18], with ongoing debate surrounding the definition of aversive techniques and whether the use of aversive-based training is justified in certain cases. While some advocate for the exclusive use of reward-based approaches without exceptions, others incorporate aversive-based techniques as a standard component of training or reserve them for specific scenarios.

The confusion for dog guardians about training methods is intensified because dog training is an unregulated profession in Canada, as in many other countries [18,19,20]. The lack of licensure allows anyone to work as a dog trainer without any specific education, skills, or requirement to apply scientifically proven methods [19]. Moreover, in unregulated fields, individuals can claim expertise without proving their knowledge or skills [20]. This enables the use of outdated methods, increases confusion, and puts the burden on guardians to educate themselves when selecting a dog trainer [19,20,21].

Guardians may have trouble evaluating trainers and training advice as the available information can be inaccurate, misleading, or incomplete (e.g., [3,4,22]). This information often comes from informal or indirect sources such as TV shows, online videos, and internet forums [21]. Moreover, it is not clear to what extent guardians understand the welfare implications of different training methods [23].

Independent third-party organizations such as professional dog training organizations, training schools, and welfare organizations have taken on the role of certifying trainers who complete specific educational requirements or fulfill certain qualifications. These credentialing organizations may issue certifications to individual trainers (e.g., the Certification Council for Professional Dog Trainers [CCPDT], International Association of Animal Behavior Consultants [IAABC], and Karen Pryor Academy). Trainers can also acquire memberships to professional associations if they meet the criteria set by the organization, such as adhering to a code of ethics or maintaining continuing education credits (e.g., the Canadian Association of Professional Dog Trainers [CAPDT], IAABC, and Pet Professional Guild). However, as noted by Foubert [19], memberships to dog trainer associations are easily obtained, in contrast to certifications, which are generally more rigorous. In Canada, there is also a non-profit auditing and accreditation program for dog training businesses operated by the British Columbia Society for the Prevention of Cruelty to Animals (BC SPCA) (AnimalKind; [24]). All these systems of voluntary self-regulation are limited, as they lack formal authority and depend on the applicants’ interest in enrolling in them [18,20]. However, they can be helpful in increasing accountability and providing a platform to connect vetted professionals with interested consumers [18]. A survey of guardians’ satisfaction with dog training schools in Australia found that client satisfaction correlated with instructors being competent and holding some form of professional accreditation, among other factors such as approachability and kindness [25].

Canadian professional dog training associations are also examining ways in which the profession can standardize or self-regulate. For example, CAPDT is working to develop a national curriculum for dog trainer education and thus set a professional standard [26]. The mission of the Professional Animal Care and Training Association of BC (PACTA BC) of the province of British Columbia (B.C.) includes the aim to “investigate whether regulation, or some form of licensing, would ensure humane treatment of animals and protect those who love and live with them” [27].

Some studies have been conducted to learn about professional dog trainers and their views. Skyner et al. [18] set out to examine the industry’s readiness for the creation of a national accreditation program in New Zealand through an online survey targeted at trainers, behavioral consultants, veterinarians, and other professionals working with dogs. They found that over 60% of the respondents were interested in accreditation, but there was a significant association between this and the respondents’ professional role. Animal trainers were the largest group that reported being uninterested or unsure about accreditation (43%, 74/143), and they were also the group that was most likely to use aversive training methods.

Recently, we collected and analyzed information from public websites of dog training businesses in B.C., Canada, to learn about the training credentials, services, modes, and methods used [28]. We identified that 72% (203/281) used only reward-based training methods, and it was more likely for reward-based training businesses than aversive-based businesses to list training credentials on their websites. Similarly, Johnson and Wynne [4] found that reward-based trainers are more likely to be certified by a third-party credentialing organization. We also found that lead trainers were more likely to be women and hold at least one training credential (consistent with the findings of [4,18,29], among others). However, this research was limited by the collection of publicly available data, which restricted interpretations in terms of self-identification and overall views of the trainers.

Therefore, in the current study, we set out to characterize the dog training profession in Canada through an online survey of individual trainers. We collected the demographic characteristics of professional dog trainers, their educational background, qualifications and training experience, the range of services they offered, terms they use to describe their training, and training items they use. We also aimed to understand their opinions on professional regulation. This research aims to contribute insights into the professional dog training sector in Canada, potentially informing best practices, educational programs, and advocacy efforts for the welfare of dogs.

## 2. Materials and Methods

This survey was aimed toward dog trainers working professionally (i.e., as a paid occupation) in Canada. Respondents were recruited through emails to dog trainers and dog training-related organizations, both in the researchers’ professional networks and those with whom the researchers did not have prior relationships (i.e., “cold calls”), as well as paid advertising on social media channels (Facebook, Instagram, and LinkedIn). To improve sample representation, we identified specific dog trainers in smaller population provinces and territories (New Brunswick, Newfoundland, Northwest Territories, Nunavut, Prince Edward Island [P.E.I.], and Yukon) to “cold call” email. During the data collection period, we monitored the geographic representation of respondents and adjusted geo-targeted digital advertising to provinces and territories with lower representation in the survey (compared to the general population; see below).

Following consent, respondents completed an online survey hosted on the Qualtrics platform. Respondents could select whether to complete the survey in English or in French.

An initial version of the survey was piloted with a sample of seven dog trainers and eight researchers to obtain feedback on the wording, structure, and overall survey design. The final survey comprised 30 questions (see Appendix A) gathering demographic information, details about dog training education and credentials, characteristics of the business and services offered, and opinions on matters such as the regulation of the dog training profession, and various dog training scenarios. Note that the wording and grouping of services and behavioral concerns (e.g., “poor manners” and “reactivity”) were based on the feedback obtained during the pilot phase. We aimed to capture industry-standard language while providing enough ambiguity to allow for flexibility in user interpretation. The order of appearance of the options was randomized for question 21 to prevent biases due to specific options always appearing first (e.g., when indicating the likelihood of using terms to describe their training methods). An open-ended, optional space was provided at the survey end for respondents to share any additional comments before submitting. All questions were optional. Note that question 9 (“How many hours of dog training continuing education courses have you taken in the last two years?”) was removed from analysis due to the unreliable results it produced, and some questions will be reported elsewhere (see Appendix A for details).

Two outliers (168 and 130 h) were excluded from responses to question 14 (“On average, how many hours of training [including preparation time and follow-up] do you carry out each week?”) as these values were deemed unrealistic. Questions 18 and 23–29 delve into trainers’ beliefs about dogs, dog training, and their training abilities, as well as scenarios outlining how they would react in different training situations, and these will be reported elsewhere.

The total number of respondents who began the survey was 1012. Data cleaning procedures included the elimination of entries with less than 85% completion (*n* = 282) and respondents who said they were not in Canada, which triggered the end of the survey (*n* = 24).

This resulted in 706 self-identified training professionals included in the analyses, and of these, 640 respondents (94.9%) completed 100% of the survey. Henceforth, when percentages are reported, they are calculated out of the total number of respondents (i.e., percentage of 706). The median time to complete the survey was 20 min 3 s. Entries below half the median completion time (i.e., 10 min/600 s, *n* = 43) were flagged for in-depth review of write-in options and internal coherence. No entries were eliminated after in-depth review. In terms of language, 677 (95.89%) respondents answered in English, while 29 (4.10%) answered in French.

In some cases, respondents selected the “other: write-in” option, and their written response described an option that was the same as one of the pre-existing listed options. These answers were manually added to the totals for the existing option that best fit their description.

Descriptive analyses were carried out with Office Excel 365, and figures were made with Canva. Associations between categorical variables were tested using Chi-square tests (α = 0.05). Two types of Chi-square tests were run. First, we assessed whether responses were dependent on target variables (e.g., province and view on regulation). Second, specific analyses explored whether the observed responses for one level of a variable differed significantly from the expected responses (e.g., differences in views on regulation among trainers from Alberta). These two tests are reported separately when appropriate, and complete results including non-significant findings are included in Appendix A. Significant findings include Cramer’s V as a measure of association between variables. Following Rea & Parker [30], these are interpreted as follows: negligible (<0.10), weak (0.10–0.20), moderate (0.20–0.40), relatively strong (0.40–0.60), strong (0.60–0.80), and very strong (>0.80).

When the assumptions of the Chi-square test were not met (i.e., expected counts were less than 5), we employed Fisher’s exact tests. In certain instances, the contingency tables for the Fisher’s exact tests exceeded the capacity of our computational workspace. In such cases, the *p*-values were estimated using Monte Carlo simulations. The number of replicates in simulation (B) was chosen as 2000, implying a minimum *p*-value of 1/(B+1) = 0.0005. These analyses were carried out with R (version 4.4.2).

A thematic analysis was conducted for some of the open-ended responses (see below) using the qualitative software package NVivo 14. CC examined the responses and created a coding framework that was used to identify key themes (detailed findings are reported in Appendix A). Responses could contain multiple themes. A second observer coded 30% of these responses to check reliability (agreement was substantial for “who should lead regulations” (k = 0.640, SE = 0.057, *p* < 0.001) and moderate for “final comments” (k = 0.433, SE = 0.050, *p* < 0.001) [31]).

To facilitate reading, the variables “generally referring to a veterinarian”, “suggesting asking a veterinarian about medication”, and “suggesting consulting with a behaviorist” from question 19 were grouped together in a new variable named “recommending veterinary consultation” for the association analyses.

## 3. Results

### 3.1. Demographic Information

Respondents were distributed across Canadian provinces as follows: B.C.: 233 (33.00%), Ontario: 221 (31.30%), Alberta: 112 (15.86%), Quebec: 55 (7.79%), Nova Scotia: 33 (4.67%), Saskatchewan: 18 (2.54%), Manitoba: 10 (1.41%), New Brunswick: 10 (1.41%), Newfoundland: 8 (1.13%), P.E.I.: 2 (0.28%), Yukon: 1 (0.14%), and Northwest Territories and Nunavut: 0; 3 respondents (0.42%) did not provide their location in Canada.

We compared these results with population estimates in Canada [32] to understand the general representativeness of the sample. Our sample may be overrepresented for B.C. and underrepresented for Quebec and Ontario. However, this comparison is approximate, as general population estimates may differ from the actual distribution of dog trainers across provinces.

Respondents were on average 42.87 years old (SD: 13.25, range: 18–83, with 53 no responses). Regarding their self-reported gender, 596 (84.41%) indicated they were women, 70 (9.9%) were men, 21 (2.9%) were non-binary, 1 (0.14%) listed “agender” as a write-in option, and 18 (2.54%) preferred not to answer or left the question blank.

When asked about their cultural background, 455 respondents (64.44%) identified as European, 38 (5.38%) as First Nations or Indigenous, 10 (1.41%) as East Asian, 3 (0.42%) as Southeast Asian, 1 (0.14%) as South Asian, 9 (1.27%) as Hispanic or Latina/o, 7 (0.99%) as Middle Eastern, and 4 (0.56%) as African. Analysis of the 86 write-in responses revealed that 60 respondents (8.49%) identified as “Canadian” and 26 (3.68%) as “Caucasian” and/or “white”, while 82 respondents (11.61%) preferred not to answer.

Regarding highest level of education, 310 respondents (43.90%) finished college or university, while 147 (20.82%) had completed some college or university. A total of 73 (10.33%) had finished graduate education, and 20 (2.83%) had some graduate education. Additionally, 47 respondents (6.67%) had completed high school, and 11 (1.55%) had completed some high school. Finally, 42 (5.94%) had completed apprenticeship training or trades, and 34 (4.81%) had a professional degree (e.g., Bachelor of Laws, Doctor of Veterinary Medicine, Medical Doctor). Ten (1.41%) respondents preferred not to answer or left the question blank.

### 3.2. Dog Training Experience and Certifications

Respondents had been training dogs professionally for an average of 10.38 years (SD: 8.88, range: 0–54, with 6 no responses). When asked how they learned to train dogs, 465 (65.86%) indicated that they had completed one or more training programs, while 235 (33.28%) indicated that they were self-educated (e.g., learned by training their own dog, reading books, watching training videos, and/or taking continuing education courses), and 6 (0.84%) left the question blank.

In terms of certifications, 209 (29.60%) indicated that they obtained their certification through graduation from a training program, 112 (15.86%) were evaluated and/or examined by a credentialing board, and 191 (27.05%) reported both. Finally, 170 (24.07%) did not have certifications, 22 (3.11%) said they were not sure, and 2 (0.28%) left the question blank.

We asked which training programs were completed, and the five most popular results were the Fear Free Animal Trainer Certification Program (80, 11.33%), Karen Pryor Academy: Certified Training Partner (KPA-CTP) (74, 10.48%), Michael Shikashio: Aggression in Dogs Master Course (77, 10.90%), Do More with Your Dog: Certified Trick Dog Instructor (CTDI) (52, 7.36%), and Animal Behavior College: Animal Behavior College Dog Trainer (ABCDT) (43, 6.09%). In terms of certificates from credentialing boards, the two most frequently selected were Certified Professional Dog Trainer-Knowledge Assessed (CPDT-KA) by the Certification Council for Professional Dog Trainers (162, 22.94%) and Certified Dog Behavior Consultant (CDBC) by the International Association of Animal Behavior Consultants (29, 4.10%). A total of 138 different training programs and 39 certifications were identified. See Table 1 and Table 2 for detailed results and Appendix A for additional write-in options added by respondents.

### 3.3. Characteristics of Dog Training Services

When asked how they were paid for their training services, 563 respondents (79.74%) indicated they were self-employed with their own dog training business, 140 (19.83%) were employees in a pet business, 71 (10.05%) were employed by a shelter or other animal non-profit organization, and 7 left the question blank. On average, the trainers reported that they spent 25 h (SD: 18, range: 0–100, with 14 non-responses and 2 excluded outliers; see the note in the Data Analysis section) training every week, including preparation time and follow-ups.

To provide training, 636 respondents (90.08%) offer in-person, one-on-one (private) sessions with the client and their dog, and 465 (65.86%) offer in-person group classes. Considering virtual options, 267 respondents (37.81%) provide online private sessions, 104 (14.73%) offer online group classes, and 71 (10.05%) offer self-paced online courses. “Day training”, where the trainer picks up the dog from the home and returns them after the session, is offered by 238 respondents (33.71%). Additionally, 166 respondents (23.51%) provide home-based board-and-train services, while 65 (9.20%) offer board-and-train in a kenneling facility. Finally, 75 respondents (10.62%) provide training in animal shelters.

Respondents provide a variety of types of training services: 615 (87.11%) offer basic training (e.g., manners, puppy classes, and recall), 556 (78.75%) work with behavior concerns (e.g., fear, reactivity, and aggression), and 310 (43.90%) offer training to address separation anxiety (presented separately from behavior concerns). Dog sports, including trick training, are offered by 332 (47.02%); 95 (13.45%) provide service dog training, and 93 (13.17%) train therapy dogs. There were 49 (6.94%) respondents who mentioned offering other services related to working dog training.

Other animal-related services provided by respondents’ training businesses included boarding services (overnight care at a facility) (156, 22.09%), pet-sitting at the client’s home (73, 10.33%), dog daycare (129, 18.27%), dog walking (165, 23.37%), retail of pet supplies (120, 16.99%), grooming (78, 11.04%), and cosmetic dental scaling (3, 0.42%). An additional 6 respondents mentioned home-based boarding in the write-in space. Concerning the training of other animals, 46 (6.51%) provided cat training and behavioral support, and 21 (2.97%) trained other species. Lastly, 274 (38.81%) indicated that the business they trained in did not offer any other services apart from dog training.

We asked respondents to select up to three of the most frequent types of training and behavioral concerns they encountered (see Figure 1). Respondents selected reactivity to dogs (455, 64.44%), poor manners (e.g., jumping up, leash pulling, mouthing) (450, 63.73%), poor recall (212, 30.02%), fear (208, 29.46%), reactivity to people (166, 23.51%), and bite history to people (83, 11.75%). Additional concerns were resource guarding (70, 9.91%), separation anxiety (64, 9.06%), barking unrelated to separation anxiety (59, 8.30%), and bite history to dogs (54, 7.64%). Other less frequent concerns were destruction/chewing/digging (without separation anxiety) (33, 4.67%) and house soiling (19, 2.69%). There were 28 respondents (3.96%) who stated that their businesses did not address any of these concerns, and 8 (1.13%) left the question blank.

Regarding non-training services or products that trainers may recommend to clients, 155 (21.95%) recommended pheromones, 124 (17.56%) massage therapy, 118 (16.71%) Tellington Touch, 107 (15.15%) cannabis products, 81 (11.47%) acupuncture, and 53 (7.50%) homeopathy. In terms of diet-related recommendations, 122 (17.28%) indicated a calming diet, 119 (16.85%) a raw diet, and 42 (5.9%) a low-protein one.

Veterinary consultations were recommended by 508 respondents (71.95%), with 336 (47.59%) specifically recommending consultations for behavioral medications and 352 (49.85%) recommending referrals to a veterinary behaviorist. Additionally, 168 respondents (23.79%) recommended spaying or neutering.

Using the write-in space, 25 (3.54%) mentioned recommending management techniques and lifestyle changes, including increasing exercise and enrichment. Additionally, 27 (3.82%) referred to diet, suggesting examining the diet themselves or consultation with a nutrition expert, while 9 (1.27%) mentioned adding probiotics or other supplements. Finally, 13 (1.84%) mentioned physical therapy, canine fitness trainers, or chiropractors, either as standalone, for pain consultation, or after veterinary consultation. There were 67 (9.49%) who indicated they did not make any non-training recommendations and 11 (1.55%) who left the question blank.

### 3.4. Training Method Descriptions

Respondents were asked how likely it would be for them to use each of these terms to describe their training methods (see Figure 2 and Appendix A). There were 625 respondents (88.52%) who indicated they were likely to use the term “positive reinforcement”, while 16 (2.26%) considered it unlikely. Using the term “reward-based” was likely for 606 respondents (85.83%) and unlikely for 15 (2.12%). When it came to “humane”, 590 respondents (83.56%) were likely to use the term, while 21 (2.97%) were unlikely to do so. “Relationship-based” was likely for 571 respondents (80.87%) and unlikely for 33 (4.67%). “Science-based” was seen as likely by 496 respondents (70.25%) and unlikely by 64 (9.06%). The term “force-free” was seen as likely by 351 respondents (49.71%) and unlikely by 190 (26.91%). For “obedience”, 341 respondents (48.30%) felt it was likely, while 158 (22.37%) found it unlikely. There were 272 respondents (38.52%) who indicated they were likely to use the term “balanced”, while this would be unlikely for 353 (50.00%). The term “pack leader” was likely to be used for 74 respondents (10.48%) and unlikely for 521 (73.79%). Lastly, only 12 respondents (1.69%) considered it likely to use the term “dominance theory”, whereas 609 (86.26%) deemed it unlikely.

Respondents could also write in other terms they used to describe their training. Remarkably, these responses were aggregated into over 80 different themes (see Appendix A). The most frequently mentioned themes included being “fear-free” (26, 3.68%), “fun”- or “games”-based (23, 4.67%), and LIMA (Least Intrusive, Minimally Aversive, i.e., using the least aversive strategies possible but using aversive techniques as a last resort, 19, 2.69%). There were 14 (1.98%) mentions related to being “positively balanced” or “positive first”, as well as 5 (0.71%) mentions of “including all quadrants”. Notably, 24 respondents (4.67%) used technical terms based on learning theory (positive/negative reinforcement/punishment, luring, capturing, etc.). Other themes included “choice” or “agency” (9, 1.28%), “consent” (8, 1.13%), and “cooperation” (8, 1.13%). Additionally, 12 respondents (1.70%) categorized their responses under the concept of “ethical”, 9 (1.28%), under “benevolent,” and 7 (0.99%) under “fair.” Finally, 14 respondents (1.98%) suggested training methodology should vary according to the characteristics of each dog (e.g., “tailored to the individual dog” and “no dog and owner combo fits one methodology”), and 10 (1.42%) referred to their training being “effective” or “results-based”.

### 3.5. Use of Different Training Items

Trainers were asked about the likelihood of using or recommending different items in their training (see Figure 3 and Figure 4, Appendix A).

In terms of collars that are typically considered aversive, 629 respondents (89.09%) indicated they would be “somewhat unlikely” or “extremely unlikely” (hereafter reported together as “unlikely”) to use or recommend a citronella collar, while 25 (3.54%) would be “somewhat likely” or “extremely likely” to do so (hereafter reported together as “likely”; for disaggregated results for “unlikely” and “likely” see Appendix A). For choke collars, 556 respondents (78.75%) were unlikely to use them, while 77 (10.91%) were likely. For electronic collars, 444 (62.89%) were unlikely to use them, while 185 (26.20%) were likely. For prong collars, 429 (60.76%) were unlikely to use them, while 214 (30.31%) were likely.

Concerning other collars that are typically not considered aversive, 466 (66.01%) were likely to use a flat collar, 317 (44.89%) a martingale collar, and 220 (31.16%) a head collar; while 82 (11.61%), 211 (29.89%), and 356 (50.42%) were unlikely to use these tools, respectively. In terms of harnesses, 367 (51.98%) were likely to use a back-clip harness, 356 (50.42%) a front-clip harness, and 186 (26.34%) a “no-pull” harness; while these were unlikely to be used by 225 (31.87%), 193 (27.33%), and 298 (42.21%), respectively.

Some items were generally likely to be used. For instance, 678 respondents (96.03%) indicated that they were likely to use food treats, 666 (94.33%) were likely to use toys, and 639 (90.51%) were likely to use a long leash (3 m/9 ft or longer), while only 7 (0.99%), 3 (0.42%), and 21 (2.97%) indicated being unlikely to use these items, respectively. Other items likely to be used included crates by 593 (83.99%), muzzles by 533 (75.50%), and clickers by 499 (70.68%), while 27 (3.82%), 45 (6.37%), and 77 (10.91%) were unlikely to use these tools.

In contrast, 566 respondents (80.17%) said it was unlikely that they would use an electronic fence, 546 (77.23%) were unlikely to use items producing startling noises (such as shaking a can with coins in it), and 585 (82.86%) were unlikely to use a water spray bottle, while it was likely that 44 (6.23%), 77 (10.91%), and 60 (8.50%) respondents would use them.

### 3.6. Opinions on Regulation

When asked whether dog training in Canada should be a regulated profession (defined as “professions requiring licenses or certifications from regulatory bodies in order to practice legally”), 439 respondents (62.18%) were in favor, 134 (18.98%) were against, 131 (18.55%) were undecided, and 2 (0.28%) left the question blank (see Figure 5). All respondents (including those who were against) were shown a follow-up question asking who should lead and create the regulations if dog training were to become regulated (see Figure 6). Using a “select all that apply” format, 369 (52.26%) indicated that “dog training professional organizations (i.e., self-regulation)” should create the regulations. Additionally, 171 (24.22%) selected “animal welfare organizations” and 98 (13.88%) chose “government agencies”, while 225 (31.86%) were “undecided regarding who should lead and create these regulations”. Out of the “undecided” responses, 161 (22.80%) chose this option exclusively, while 64 (9.06%) selected both “undecided” and one or more of the other options.

There were 163 (23.08%) respondents who selected “other” and wrote in their responses for who should lead regulations. These were analyzed using qualitative coding to identify the most prevalent themes (see Appendix A). Among these, 35 respondents (4.96%) expressed a need for collaboration, either among “all of the above” (i.e., dog training organizations, welfare organizations, and government agencies) or any combination of these, with the addition of veterinary organizations or researchers. Just nine respondents (1.28%) proposed that a novel overseeing body or council with regulatory powers be created for this purpose.

There were 34 respondents (4.81%) who wrote in responses related to “self-regulation”. For example, 15 (2.12%) referred to the need to consult “experienced trainers” such as trainers with many years of experience competing in dog sports, sport titles, or who had “proven their ability” (e.g., “dog trainers who have proven their ability across hundreds or thousands of dogs, have titled dogs, and have a body of work that supports their desire to have an opinion that matters”; “… choosing the best in their field (world champions or renowned trainers) they have already proven themselves”). There were 29 (4.11%) respondents who stated any regulation needed representation from different types of training methods. Some of the ways in which this idea was expressed included “all methods”, “all areas of dog training”, “both sides of the spectrum”, “nonbiased”, “variety of approaches”, and “balanced and force-free trainers”.

Some respondents (23, 3.25%) shared their opposition to regulations, while others (29, 4.10%) identified organizations that should not be regulators, with 16 (2.27%) opposing animal welfare organizations in this role. Respondents viewed these organizations as biased and/or that they “did not understand” dog training well enough to regulate it. Additionally, 9 respondents (1.28%) mentioned concerns about the government overseeing regulations. Finally, there were 9 (1.27%) mentions of themes related to concerns that regulations would create barriers to entry into the profession for systematically excluded groups, such as the cost of licensing.

### 3.7. Final Comments

The final question invited respondents to share any additional comments they had about dog training in a write-in text box and was completed by 292 (41.1%) respondents (see Appendix A). Responses were grouped into four main themes: comments on specific dog training methods, regulations, concerns about the industry, and feedback about the survey itself.

Some (148, 20.96%) respondents used the open comments to express their support or disagreement with specific training practices. As such, some advocated for banning aversive-based tools such as prong collars, electric collars, and electric fences, citing examples from countries where such bans are in effect. In contrast, others supported the use of aversive tools and sometimes indicated that only using positive methods would result in more euthanasia (e.g., “I would rather use a prong or an ecollar [sic] on a dog that needs it if this will stop them from being euthanized or in a shelter…”). Additionally, some commented on the usefulness and necessity of having the option of balanced training. Some shared their worries about harm to dog welfare from aversive training methods. Conversely, others were concerned that reward-based training may create long-term behavioral issues resulting in behavioral euthanasia. Interestingly, trainers from either of these opposing views said they had observed an increase in the number of dogs where the methods used by a previous trainer “created” or made a behavior issue “worse”. Finally, some respondents said that dog training was not “one size fits all”, often coupled with the need to maintain an “open mind” and “freedom of choice” in methodologies.

Some respondents (110, 15.58%) commented on the topic of regulation. Some of these expressed their support without conditions, underscoring the need for the field to be regulated. A small proportion of these specified that they would only favor regulation if “all [training] methods” were included. Other respondents expressed their opposition, citing concerns such as regulations that limit the use of aversive-based tools or make the profession less accessible due to increased costs, as well as the difficulty of enforcing regulations.

Other concerns about the industry were raised by 90 (12.74%) respondents. These included worries about the polarization between reward-based and aversive-based approaches as well as the potential for guardians to become discouraged or confused by the variability in approaches and the use of misleading language. Other areas of concern included the lack of trainer education and ease of obtaining online certifications with minimal qualifications, the risk of dogs developing issues due to receiving “bad” training, and concerns about the cost and accessibility of training. Finally, some highlighted the need for more education either for trainers, guardians, or both, as well as the importance of practical experience for trainers.

There were 50 (7.08%) respondents who commented about the survey itself. Among these, some expressed concerns about the perceived bias of the survey against balanced trainers, and/or were critical of the wording of the questions. Specifically, the lack of detail in the response options, the belief that incorrect terminology was used in questions, and that more consultation with trainers was needed to improve questions.

Finally, there were 28 (3.96%) responses which were grouped as “other comments”. These included statements that did not fit in the other themes, indications of appreciation for conducting the survey, and clarifications of the answers to other questions.

### 3.8. Associations with Provinces

Associations with provinces were explored in depth to better understand regional differences in dog trainers across Canada. These analyses were only calculated for Alberta, B.C., Ontario, and Quebec, as these provinces held the largest number of respondents (*n* = 621, 87.96% of all responses), and more consistently fulfilled the requirements for Chi-square tests (see Appendix A). Note that there were no significant associations for the analyzed variables for trainers in B.C.

There were significant associations of province with trainer opinion on regulation (X^2^ (6, *N =* 619) = 14.72, *p* = 0.023; Cramer’s V = 0.11). When comparing results between provinces, it appears that it was more likely than expected for trainers in Alberta and Quebec to be in favor of regulation, while it was less likely than expected for trainers in Ontario (however, note that tests within each province were not significant, ps > 0.069, so this is merely a descriptive analysis of these observations). On the other hand, there were no significant associations between province and believing that if dog training were to be regulated, this should be led by welfare organizations (X^2^ (3, *N* = 615) = 6.37, *p* = 0.095), dog training organizations (X^2^ (3, *N* = 615) = 3.36, *p* = 0.34), or the government (X^2^ (3, *N* = 615) = 6.33, *p* = 0.097). When comparing results within each province, it was found that trainers in Alberta were more likely than expected to be in favor of welfare organizations taking on this role (X^2^ (1, *N* = 111) = 3.89, *p* = 0.048, Cramer’s V = 0.19), while in Quebec, trainers were more likely than expected to support the government for this role (X^2^ (1, *N* = 55) = 4.45, *p* = 0.035, Cramer’s V = 0.28).

Province was also significantly associated with how respondents learned to train dogs (X^2^ (3, *N* = 618) = 8.8, *p* = 0.032; Cramer’s V = 0.12). Specifically, trainers in Quebec were less likely than expected to be self-educated (X^2^ (1, *N* = 55) = 5.05, *p* = 0.025, Cramer’s V = 0.3). However, there were no significant associations of province with whether trainers were certified through a training program, credentialing board, or both (X^2^ (9, *N* = 603) = 12.13, *p* = 0.206). When comparing results within provinces, it was observed that in Alberta, trainers were more likely than expected to have certifications from both a training program and a credentialing board (X^2^ (3, *N* = 111) = 8.38, *p* = 0.039, Cramer’s V = 0.16).

Regarding the terms used to describe their training methods, there was a significant association between province and the likelihood of using the term “balanced” (X^2^ (6, *N* = 610) = 18.47, *p* = 0.005; Cramer’s V = 0.12). The use of “balanced” was more likely than expected for trainers from Ontario (X^2^ (2, *N* = 218) = 8.08, *p* = 0.018), while trainers in Alberta indicated being unlikely to use this term at a rate higher than expected (X^2^ (2, N = 112) = 6.42, *p* = 0.04). There were also significant differences in the use of the term “obedience” across provinces (X^2^ (6, *N* = 612) = 22.54, *p* = 0.001; Cramer’s V = 0.14), as Ontario trainers were more likely than expected to use ”obedience” (X^2^ (2, *N* = 219) = 8.89, *p* = 0.012, Cramer’s V = 0.14), while the opposite occurred in Quebec (X^2^ (2, *N* = 55) = 10.21, *p* = 0.006, Cramer’s V = 0.3). Similarly, “pack leader” had a significant association between province and the likelihood of using the term (X^2^ (6, *N* = 609) = 19.49, *p* = 0.003; Cramer’s V = 0.13). Trainers in Ontario were more likely than expected to describe themselves as “pack leader” (X^2^ (2, *N* = 219) = 8.77, *p* = 0.012, Cramer’s V = 0.14), while trainers in Alberta were less likely to do so (X^2^ (2, *N* = 111) = 6.92, *p* = 0.031, Cramer’s V = 0.14).

There were also associations for the term “relationship-based” (Fisher’s exact test, simulated *p*-value: *p* = 0.032), which was more likely to be used by trainers from Ontario (X^2^ (2, *N* = 219) = 6.27, *p* = 0.043, Cramer’s V = 0.12). There were also differences in the use of “force-free” (X^2^ (6, *N* = 610) = 30.65, *p* < 0.001; Cramer’s V = 0.16), where trainers from Quebec were more likely than expected to use it (X^2^ (2, *N* = 55) = 16.29, *p* < 0.001, Cramer’s V = 0.38), and trainers from Ontario were less likely (X^2^ (2, *N* = 219) = 8.6, *p* = 0.014, Cramer’s V = 0.14). No significant associations were found between provinces and the terms “dominance theory”, “humane”, “positive reinforcement”, “science-based”, and “reward-based” (Fisher’s exact test, simulated *p*-value, *ps* > 0.115).

Considering the use of collars that are typically considered aversive, significant differences were found for trainers who indicated that they were likely to use e-collars (Fisher’s exact test, simulated *p*-value: *p* < 0.001), prong collars (Fisher’s exact test, simulated *p*-value: *p* < 0.001), and choke collars (Fisher’s exact test, simulated *p*-value: *p* = 0.038). For all of these, it was more likely than expected that a trainer who was likely to use these collars would be located in Ontario (*ps* < 0.025, for complete results, see Appendix A).

### 3.9. Associations with Trainer Opinions on Regulation

There was a statistically significant association between the trainer’s views on whether the dog training profession should be regulated and how they learned to train dogs (X^2^ (2, *N* = 698) = 50.68, *p* < 0.001, Cramer’s V = 0.27) (see Table 3 and Appendix A). It was more likely than expected that a trainer would be against (X^2^ (1, *N* = 132) = 25.76, *p* < 0.001, Cramer’s V = 0.44) or undecided about (X^2^ (1, *N* = 130) = 6.98, *p* = 0.008, Cramer’s V = 0.23) the possibility of regulations if they were self-educated. On the contrary, it was more likely than expected that they would be in favor (X^2^ (1, *N* = 436) = 17.94, *p* < 0.001, Cramer’s V = 0.2) of regulation if they had completed one or more training programs.

Similarly, there was an association between the opinion on regulations and holding dog training certifications (X^2^ (6, *N* = 680) = 73.95, *p* < 0.001, Cramer’s V = 0.23). It was more likely than expected that a trainer would be against (X^2^ (3, *N* = 127) = 43.4, *p* < 0.001, Cramer’s V = 0.34) regulation if they did not hold any certifications. Conversely, it was more likely than expected they would be in favor of regulation if they had received certificates from a credentialing board (while this was not observed in the case of training school credentials) (X^2^ (3, *N* = 427) = 24.26, *p* < 0.001, Cramer’s V = 0.14).

Additionally, there were associations between general opinion on regulation and believing that regulation should be led by animal welfare organizations (X^2^ (2, *N* = 697) = 83.56, *p* < 0.001, Cramer’s V = 0.35) as well as by regulation being led by the government (X^2^ (2, *N* = 697) = 41.06, *p* < 0.001, Cramer’s V = 0.24). In both cases, trainers who were against regulation were also less likely than expected to support these organizations overseeing the regulatory process (welfare organizations: X^2^ (1, *N* = 130) = 37.13, *p* < 0.001, Cramer’s V = 0.53; government: X^2^ (1, *N* = 130) = 12.98, *p* < 0.001, Cramer’s V = 0.32). In contrast, if trainers were in favor of regulation, they were more likely to support these organizations taking a leading role (welfare organizations: X^2^ (1, *N* = 438) = 30.27, *p* < 0.001, Cramer’s V = 0.26; government: X^2^ (1, *N* = 438) = 15.26, *p* < 0.001, Cramer’s V = 0.19). There were no significant associations between opinion on regulation and the process being led by dog training organizations (self-regulation) (X^2^ (2, *N* = 679) = 4.29, *p* = 0.117).

Views on regulation were also associated with the use of certain training method terminology. In the case of the term “balanced” (X^2^ (4, *N* = 693) = 191.13, *p* < 0.001, Cramer’s V = 0.37), trainers were more likely to be against regulation if they described their training using this word (X^2^ (2, *N* = 130) = 103.26, *p* < 0.001, Cramer’s V = 0.63). Similar findings were observed in the case of “obedience” (X^2^ (4, *N* = 694) = 96.47, *p* < 0.001, Cramer’s V = 0.26) and “pack leader” (X^2^ (4, *N* = 691) = 91.04, *p* < 0.001, Cramer’s V = 0.25), as trainers who were likely to use these terms were also more likely than expected to be against regulation (obedience: X^2^ (2, *N* = 129) = 55.1, *p* < 0.001, Cramer’s V = 0.46; pack leader: X^2^ (2, *N* = 129) = 58.03, *p* < 0.001, Cramer’s V = 0.47). Additionally, in the case of “force-free” (X^2^ (4, *N* = 692) = 196.54, *p* < 0.001, Cramer’s V = 0.38) and “science-based” (X^2^ (4, *N* = 692) = 112, *p* < 0.001, Cramer’s V = 0.28), it was observed that trainers who indicated that they were unlikely to use these terms were more likely than expected to be against regulation (force-free: X^2^ (2, *N* = 129) = 115.55, *p* < 0.001, Cramer’s V = 0.67; science-based: X^2^ (2, *N* = 129) = 49.17, *p* < 0.001, Cramer’s V = 0.44). Conversely, there were associations with the view on regulation and the use of the terms “humane” (Fisher’s exact test, *p* < 0.001), “reward-based” (Fisher’s exact test, *p* = 0.001), “positive reinforcement” (Fisher’s exact test, *p* < 0.001), and “science-based” (X^2^ (4, *N* = 692) = 112, *p* < 0.001, Cramer’s V = 0.28). Descriptively, trainers who were likely to use these terms were also more likely than expected to be in favor of regulation, but most of the specific comparisons did not fulfill the testing criteria or did not reach significance (see Appendix A).

Finally, there were associations between the view on regulation and the use of collars that are typically considered aversive. Trainers who were against regulation were more likely than expected to use e-collars (X^2^ (2, *N* = 130) = 142.16, *p* < 0.001, Cramer’s V = 0.74), prong collars (X^2^ (2, *N* = 128) = 152.95, *p* < 0.001, Cramer’s V = 0.77) and choke collars (X^2^ (2, *N* = 129) = 49.24, *p* < 0.001, Cramer’s V = 0.44).

### 3.10. Associations with Recommending Veterinary Consultation

Inspired by Hunter et al.’s [5] finding that balanced trainers were less likely than reward-based trainers to refer clients to veterinarians, we examined any associations with trainers recommending veterinary consultations (see Table 4, Appendix A).

When examining factors associated with increased likelihood of recommending veterinary consultation, we found that it was more likely that a trainer would recommend consultation if they were in favor of regulation (X^2^ (1, *N* = 436) = 23.09, *p* < 0.001, Cramer’s V = 0.23) and if they were located in Alberta (X^2^ (1, *N* = 112) = 12.86, *p* < 0.001, Cramer’s V = 0.27).

Conversely, we also observed some associations with not being likely to recommend veterinary consultations. Trainers who were likely to use the terms “balanced” (X^2^ (2, *N* = 132) = 96.6, *p* < 0.001, Cramer’s V = 0.6), “obedience” (X^2^ (2, *N* = 134) = 48.79, *p* < 0.001, Cramer’s V = 0.43), and “pack leader” (X^2^ (2, *N* = 132) = 76.83, *p* < 0.001, Cramer’s V = 0.54), were less likely than expected to recommend that clients consult with veterinarians. On the contrary, those who indicated that they were likely to use the terms “force-free” (X^2^ (2, *N* = 132) = 75.21, *p* < 0.001 Cramer’s V = 0.53) and “science-based” (X^2^ (2, *N* = 133) = 43.19, *p* < 0.001, Cramer’s V = 0.4) were less likely than expected to not recommend veterinary consultations.

Moreover, trainers who indicated that they were against regulation (X^2^ (1, *N* = 130) = 62.81, *p* < 0.001, Cramer’s V = 0.7) and trainers who were located in Ontario (X^2^ (1, *N* = 219) = 13.66, *p* < 0.001, Cramer’s V = 0.25) were also more likely than expected to avoid recommending veterinary consultations.

## 4. Discussion

We aimed to deepen our understanding of the demographics, education, training methods, and views of professional dog trainers in Canada based on the survey responses of individual trainers.

The surveyed trainers predominantly consisted of women of European descent with an average age of 43 and with college degrees or some college education. The majority (80%) of trainers indicated being self-employed and working an average of 25 h per week. This characterization of the Canadian dog training sector is similar to previous research, in which women have been reported to outnumber men in this profession [18,28,29]. Notably, the use of self-identification in this study overcomes the limitations of prior research in which the gender of trainers was inferred [4,28]. Regarding the highest level of non-training education, our findings are consistent with those of Skyner et al. [18], in which 40% of the respondents held a tertiary certificate or diploma and 22% had obtained a bachelor’s degree.

In terms of cultural background, 64% of respondents identified as European, with the second-most frequent option being the write-in addition of “Canadian”, which was mentioned by 8%, and First Nations or Indigenous, which was selected by 5%. Less than 5% identified themselves as from a non-Indigenous, non-white cultural group, while 12% preferred not to answer. There is limited research exploring cultural barriers to pursuing careers in animal care professions; however, new studies in veterinary medicine may be broadly applicable to the wider animal care sector, including training. Gordon [33] recently indicated that inequities in veterinary medicine stem from broader social inequities and threaten access to services. Moreover, understanding inequities and barriers has been identified as a prerequisite to promoting equity-focused practices [33]. In line with this, Martinez [34] reported a significant lack of racial diversity in veterinary medicine. Some of the identified systemic barriers affecting the access of marginalized communities to the veterinary profession included the high cost of education, historical injustices, and exclusionary laws. Various survey studies have also reported a higher prevalence of pet ownership in white participants [35,36,37] compared with other races and ethnicities, and Ly et al. [38] found that racialized and immigrant populations may face adversities that increase the likelihood of surrendering companion animals. Gordon [33] highlighted the importance of empathy, emphasizing that professionals who identify as part of systematically excluded groups can help break down barriers and enhance access to care. Further research is needed to understand the challenges that ethnicity and race present to pursuing a career as a dog training professional and accessing the services of a trainer, as well as to explore ways of building equitable practices in dog training.

The majority (80%) of trainers indicated being self-employed and working an average of 25 h per week. Exploration of why dog trainers appear to work part-time in Canada and whether this is a deliberate choice or a result of other factors would require additional research. For example, trainers may choose to work part-time due to holding other jobs and/or the difficulty of attracting clients. It is also possible that dog training is such a labor-intensive occupation that many dog trainers do not have the capacity to work more than part-time.

Shifting the focus to experience and qualifications, respondents reported an average of 10 years working professionally as trainers. Most (66%) had completed at least one structured training program, while 33% were self-educated. Remarkably, the trainers named 138 different training programs and 39 certifications. Our findings on trainer education further highlight and quantify that there is no standardized curriculum, educational pathway, or testing process that is followed by trainers in Canada. As mentioned, there is no legally mandated level of skill or educational qualification to work as a dog trainer [19,20]. It is thus noteworthy that the 138 training programs differed considerably in their durations and scope, with some comprising full, multi-course training programs and others being short-term programs with a narrow, single-topic focus (e.g., scent work, separation anxiety, or aggression). Organizations providing the programs were also quite different and included universities, large “corporations” educating hundreds of students yearly with online courses, and small, local schools using an apprenticeship-style model with a few students per year. Organizations providing the 39 certifications were similarly varied, including both non-profit, board-run, dog training professional organizations with international membership and small, local, privately-operated organizations (some of these may be better classified as training programs; however, we report them in the same way as the survey respondents).

Furthermore, Cavalli et al. [28] found that some trainers indicated on their websites that they were “professional” or “credentialed” without referencing specific programs. Adding to this issue, this survey identified over 80 different themes or ways in which trainers described their training practices. This incredible variability may create confusion for both dog guardians and aspiring trainers, who do not have the knowledge to identify adequate programs and credentials or parse training terminology. Navigating available training options can be challenging for guardians who may wish to review credentialing organizations to assess how training methodology is described by the organization and to verify that a trainer is listed as a graduate or certificate-holder.

In terms of training delivery mode, in-person training sessions were the most popular, either one-on-one (90%) or in groups (66%). This finding differs from Skyner et al. [18], who reported that 74% of New Zealand trainers teach group classes and 54% teach one-on-one. It is unclear whether this is due to differences between Canadian and New Zealand samples or other factors that impact willingness to attend group classes, such as the COVID-19 pandemic or travel distances. The types of sessions offered may also be related to the training needs in a particular area, as dog-aggressive or fearful dogs may not be suitable for group environments. Most sessions being one-on-one and in-person could also add to the perceived limits in the capacity of trainers for taking on clients, as they may require additional time to commute to different locations. Online training options were offered by 38% of the respondents, which is similar to our finding that 38% of the websites of B.C. trainers provide online sessions [28]. The learning outcomes of online training should be further explored, as it could offer a valuable resource for guardians who might not otherwise have access to professional dog training. Also, in line with Cavalli et al. [28], 34% provided “day training”, and 33% offered board-and-train either at home or in a kenneling facility.

The most offered type of training service was basic training (87%), followed by training for behavior concerns (79%). Trainers said the top three most common training concerns were reactivity to dogs (64%); poor manners, including jumping, leash pulling, and mouthing (64%); and poor recall (30%). This aligns with prior studies indicating poor manners as some of the most frequently guardian-reported problems, including overexcitement, pulling on the lead, jumping up at people, and poor recall (e.g., [9,22,39]). Moreover, behavioral problems have been reported as a cause for the relinquishment of dogs to shelters (e.g., [40,41]), highlighting the importance of addressing these issues and providing support to guardians before they become unmanageable. In particular, aggressive behavior has been identified as a reason for relinquishment (e.g., [42,43]).

To learn about training methods used, we asked about the likelihood of using certain terms to describe training methods and of using certain training items. An overwhelming majority were likely to use food treats (96%) and toys (94%), suggesting that dog trainers in Canada are well aware of the evidence showing that dog learning is motivated by the use of rewards, even if the trainer also incorporates the use of aversives (i.e., balanced training). Moreover, the majority were also likely to use terms associated with reward-based training and indicated they would be unlikely to use collars typically considered aversive.

Taken together, these findings suggest that reward-based training may be the most prevalent training method in Canada (but see Limitations below). Furthermore, in line with this, most training programs and credentials identified by respondents were from organizations promoting solely reward-based training or LIMA approaches, which encourage the use of Least Intrusive Minimally Aversive strategies but do not completely exclude the use of aversive techniques [44]. These results align with our previous finding that 72% of training businesses in B.C. used reward-based training methods [28]. However, as this is the first time these figures have been reported, there is not enough evidence yet to conclude whether these numbers are trending in one direction or remaining static over time.

We explored whether the likelihood of using certain terms or tools differed geographically. We observed that trainers in Ontario were more likely than expected to use the terms “balanced”, “obedience”, and “pack leader” and to use or recommend electronic and prong collars, all of which are typically considered aversive. However, Ontario trainers were also more likely to use the term “relationship-based”, which is not typically associated with aversive-based training. Trainers from Alberta and Quebec were more likely than expected to use terms generally associated with reward-based training and less likely to use collars typically considered aversive. Nevertheless, interpretation based on these data should be made with caution, as the number of respondents from some of the provinces limited these analyses. Future studies would be needed to expand and confirm this finding and explore possible reasons.

Trainers were asked about other services or products beyond training they may recommend, including veterinary consultations. Interestingly, some trainers added in the write-in space that they typically request a veterinary consultation before starting to work with a client in order to rule out undiagnosed pain or other medical problems. Conversely, others explained they involved veterinarians only as a “last resort” if training did not improve the dog’s behavior.

Hunter et al. [5] examined the beliefs and management of canine separation anxiety of Australian reward-based and balanced trainers, who differed significantly in their views on veterinary involvement and the use of medication. Balanced trainers were less likely to refer clients to veterinarians and believed medication was rarely needed compared to reward-based trainers, who rated assistance from a veterinary behaviorist as extremely important in effectively managing separation anxiety and were likely to recommend that dog guardians ask their veterinarian for medication. Additionally, balanced trainers were more likely to consider separation anxiety as a preventable issue. The authors suggested that balanced trainers may be more inclined to believe that separation anxiety can be addressed through changes in human behavior, reducing the need for pharmacological intervention. In line with this, we found that trainers who were likely to use the terms “balanced”, “pack leader”, and “obedience” to describe their training were less likely than expected to refer clients to a veterinarian. The opposite was observed in the case of trainers unlikely to use the term “dominance theory” and likely to use the terms “force-free”, “science-based”, “positive reinforcement”, and “reward-based”, as they were more likely than expected to suggest veterinary involvement. Hunter et al. [5] also found that guardians are likely to be guided by their trainer’s beliefs, and they may be reluctant to seek veterinary treatment if the trainer discourages it.

Additionally, Hunter et al. [5] remarked on the need for collaborative care between trainers, veterinarians, and other animal behavior professionals to effectively support dogs and guardians. However, survey responses in this research indicated that a minority of professional dog trainers may lack clarity regarding the scope of practice for trainers, as some of the non-training services or products they may recommend to clients fall under the areas of authorized practice for licensed veterinarians. For example, non-veterinarians making recommendations for the treatment of animals to prevent or correct disease, injury, defect, disorder, or a similar condition is considered “unauthorized practice” in veterinary medicine (e.g., [45]). In addition, despite Canada allowing medicinal and recreational use of cannabis products in humans, veterinarians cannot authorize the use of cannabis for their patients under the current regulations [46]. As such, recommending these products for animal consumption is not approved for any other canine professionals, including trainers. In terms of diet, raw-based diets are increasing in popularity, despite veterinary concerns due to their associated risks and limited confirmed benefits (e.g., [47,48]).

When asked about whether dog training in Canada should be a regulated profession, 62% of respondents expressed being in favor, 19% were against, and 19% were undecided. The wording of the question included both the terms “licensure” and “certification” to describe professional regulation with the aim of obtaining a general understanding of their opinion without delving into the distinction between voluntary and mandatory practices (e.g., [19]). The findings suggest that the overall opinion trends toward supporting some sort of regulation.

This is similar to Skyner et al. [18]’s findings in New Zealand, where over 60% of respondents were “interested” in accreditation. The authors noted that animal trainer respondents (as opposed to behavioral consultants, dog safety educators, veterinarians, and veterinary nurses) were the largest group that was uninterested or unsure about accreditation, and it was also the group most likely to engage in the use of aversive training methods. In the same line, we found that trainers were more likely to be against regulation if they used terms like “balanced”, “obedience”, or “pack leader” to describe their training, and if they indicated that they were extremely likely to use collars that are typically considered aversive. Conversely, trainers likely to use terms generally associated with reward-based training were more likely to be in favor of regulation.

Skyner et al. [18] also reported that most trainers who were members of animal training clubs or organizations supported accreditation. In our study, we found that it was more likely than expected that a trainer would be against or undecided about the possibility of regulation if they were self-educated. In contrast, it was more likely than expected that they would be in favor of regulation if they had completed one or more training programs and if they had dog training certificates. Prior research [4,28] has found that reward-based trainers are more likely to be certified or list credentials on their websites.

Further research is needed to clarify whether the likelihood of supporting professional regulation is influenced more by the methods employed by trainers or by their education and certification, as well as the interplay between these variables. For instance, credentialed trainers may support regulation because their certification process already includes an evaluation of their training, which could make them less concerned about having to undergo some sort of evaluation for regulatory purposes. Alternatively, certified trainers are more likely to use reward-based methods, and such trainers may have a stronger interest in limiting the use of aversives. This could lead them to support regulatory measures as a way to discourage the use of aversive-based methods. Conversely, non-certified trainers, and those who regularly use aversive-based methods, may be less supportive of regulation if they feel it would impose external controls that challenge their current practices and the viability of their training business.

We also asked for respondents’ views on who should lead and create the regulations, and 52% indicated it should be “dog training professional organizations (i.e., self-regulation)”. Moreover, those using the write-in space identified variations of “self-regulation” (5%), and some highlighted the need to consult with experienced or proven trainers (2%) and the importance of representation of trainers from “all methods” (4%). A further 24% selected “animal welfare organizations”, and 14% chose “government agencies”, although other respondents expressed concerns about these types of organizations leading regulation due to bias or limited expertise. Finally, 32% were undecided regarding who should lead and create these regulations, and some expressed worries that regulations could create barriers to entering the profession due to factors such as increased costs. Together, these findings highlight that although overall opinion trends toward supporting regulation, close collaboration among relevant organizations will be essential to create policies that reflect the realities of the profession and to obtain legitimacy and trust from trainers. Professional standardization and/or regulation would benefit dog welfare by providing assurances to dog guardians and promoting science-based, effective training methods as the societal norm while discouraging the instruction and use of harmful techniques.

Further research should also explore the motivations and challenges that influence method selection, particularly for those who use aversive-based techniques. This could provide valuable insights into how to encourage the adoption of humane practices. Perceptions of what is humane may be influenced by ethical, individual, and cultural values regarding dogs and how they learn, and exploring these perspectives may lead to a more nuanced understanding of dog welfare.

One notable limitation of this research is the possibility of bias due to self-selection of the sample. Some trainers may have refrained from participating after learning that the research team was affiliated with a well-known animal welfare organization that operates an accreditation program for reward-based trainers. As such, our results may underrepresent perspectives with opposing views.

In line with this, although the research team tried to avoid biases in the wording of the survey, no balanced trainers provided feedback during survey development. This could have led to unintentional biases persisting in the questions and available options, which may have made trainers with these views uncomfortable and/or unlikely to finish the survey.

Another limitation is related to the geographical representativeness of the sample. The research team made an effort to reach respondents from all territories within Canada through geo-targeted advertising and “cold call” emails directed toward trainers in less-represented areas. Nevertheless, the final sample appears to be overrepresented for British Columbia and underrepresented for Quebec and Ontario. As mentioned above, this analysis should be interpreted with caution because it is based on approximate population estimates and not the actual distribution of dog trainers in Canada. It would be valuable to obtain some general estimates of the number of individuals training dogs professionally in Canada.

Finally, some questions may have introduced biases or confused the respondents. For instance, asking about separation anxiety as its own category in question 17 may have artificially increased the number of trainers working with this behavioral concern. Additionally, in question 19, pooling all non-training recommendations including veterinary consultations may have resulted in an artificially low number of respondents indicating they do not offer non-training advice. Having a better estimate of this number would have been beneficial to inform the discussion of professional boundaries.

## 5. Conclusions

This research provides a starting point to further characterize the professional dog training sector in Canada and continue examining the methods and views of dog trainers. Our findings on trainer education confirm that there is no standardized curriculum, educational pathway, or testing process followed by trainers in Canada. Notably, the training terminology used by the majority of respondents suggests that reward-based training may be the most prevalent training method in Canada, although there seem to be regional variations in the prevalence of training methods. Lastly, a small majority of respondents support regulation for dog trainers. However, the wide variety of training programs, certifications, and terminology used to describe their approaches highlight the fragmented nature of the dog training field. Similarly, the lack of standardized training methods, trainer evaluation criteria, and absence of professional regulation present significant challenges for dog guardians in selecting trainers, which can result in the use of training methods that harm dog welfare. Therefore, establishing regulations and/or a standardized curriculum and evaluation for the dog training profession are promising initiatives that could bring clarity to both professionals and guardians. While it is encouraging that the majority of our sample supported regulation, the success of such initiatives depends on industry-wide acceptance. To achieve this, further research is essential to take into consideration the perspectives of various types of trainers in order to inform best practices, develop effective educational initiatives, and support animal welfare advocacy.

Some of the areas that warrant further exploration include identifying the essential curriculum required to become a dog trainer, which could be initiated by comparing the topics and learning outcomes of existing programs. Moreover, research should examine the motivations and challenges influencing method selection, particularly among trainers who use aversive-based techniques. In addition, it is crucial to investigate the role of systemic barriers, such as those related to ethnicity and race, which limit access to careers in dog training and to obtaining dog training services. This would help identify ways to address or mitigate these challenges and promote greater inclusivity within the profession. Finally, another fruitful area of study involves examining the relationship between trainers and other animal care professionals, including veterinarians and veterinary behaviorists.

## Figures and Tables

**Figure 1 animals-15-01255-f001:**
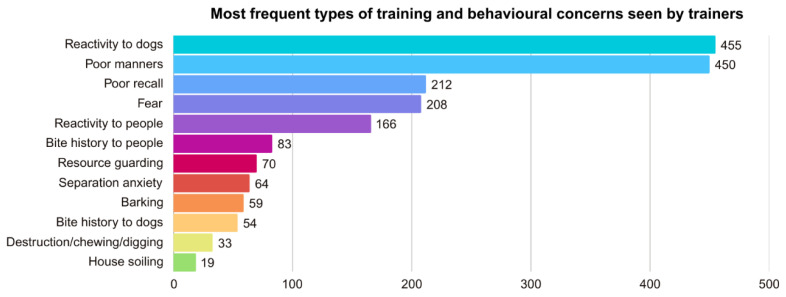
Most frequent types of training and behavioral concerns seen by trainers. Note. In the survey question, the options “barking” and “destruction/chewing/digging” were specified “without separation anxiety” to differentiate these issues as occurring independently from separation anxiety, which was included as a standalone category.

**Figure 2 animals-15-01255-f002:**
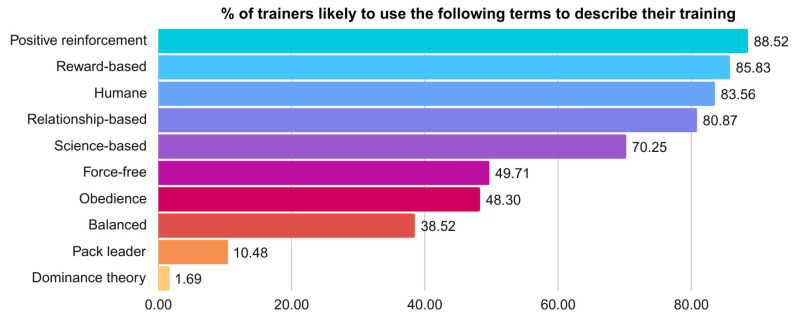
Percentage of trainers likely to use the following terms to describe their training.

**Figure 3 animals-15-01255-f003:**
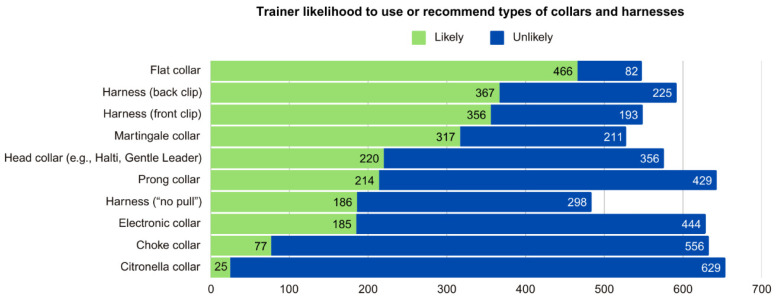
Trainer likelihood to use or recommend types of collars and harnesses.

**Figure 4 animals-15-01255-f004:**
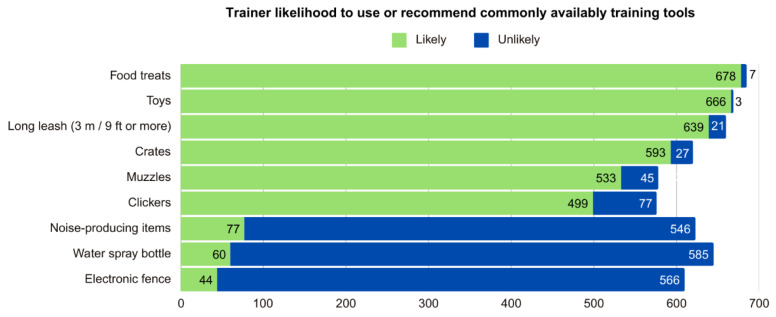
Trainer likelihood to use or recommend commonly available training tools.

**Figure 5 animals-15-01255-f005:**
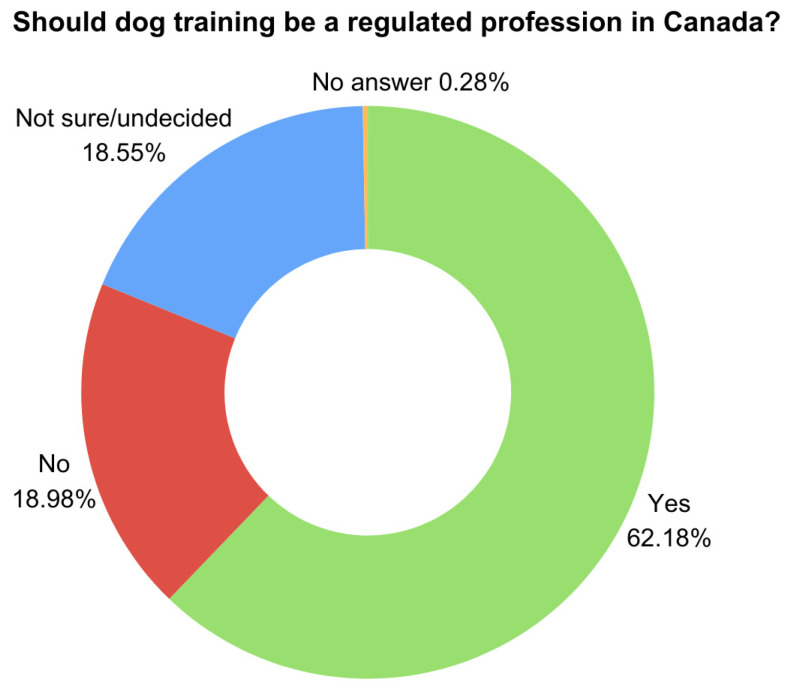
Should dog training be a regulated profession in Canada?

**Figure 6 animals-15-01255-f006:**
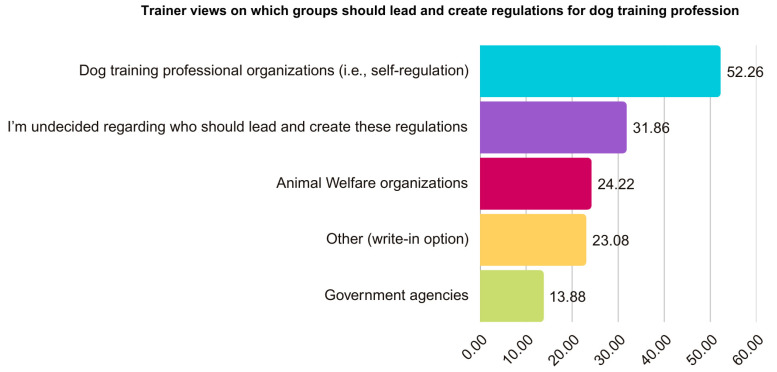
Trainer views on which groups should lead and create regulations for dog training profession. Note. This question used an “all that apply” format, so totals do not add up to 100%.

**Table 1 animals-15-01255-t001:** Training programs identified by 1% or more of respondents as a follow-up to the question “Have you obtained any dog training certifications?”.

Organization	Name of Credential	Number of Respondents
Absolute Dog Academy	Pro Dog Trainer course	27 (3.8%)
Academy for Dog Trainers	Certificate in Training and Counselling (CTC)	30 (4.24%)
Animal Behavior College	Animal Behavior College Dog Trainer (ABCDT)	43 (6.09%)
Behavior Works (Dr. Susan Friedman) ^a^	Living & Learning with Animals (LLA) professional	9 (1.27%)
Canada West Canine Training	Master Trainer	17 (2.40%)
Certified Separation Anxiety Trainer	Certified Separation Anxiety Trainer (CSAT)	16 (2.26%)
Companion Animal Sciences Institute	Diploma of Canine Behavior Science and Technology (Dip. CBST)	17 (2.40%)
Do More with Your Dog	Certified Trick Dog Instructor (CTDI)	52 (7.36%)
Dogma Academy ^a^	Dog Trainer (DDT)/Dog Behaviour Consultant (DCBC)	9 (1.27%)
Education Center	Family Dog Mediator (FDM)	37 (5.24%)
Fear Free	Fear Free Animal Trainer Certification Program	80 (11.33%)
Good Dog Academy ^a^	Professional Pet Dog Trainer (PPDT)	10 (1.41%)
Grisha Stewart Academy	Certified BAT Instructor (CBATI)	16 (2.26%)
Karen Pryor Academy	Certified Training Partner (KPA-CTP)	74 (10.48%)
Karen Pryor Academy	Dog Trainer Professional (KPA-DTP)	25 (3.54%)
Michael Shikashio	Aggression in Dogs Master Course	77 (10.90%)
National Association of Canine Scent Work	Certified Nose Work Instructor (CNWI)	12 (1.69%)
SA Pro	Julie Naismith’s Separation Anxiety Pro	14 (1.98%)
Victoria Stilwell Academy for Dog Training	Victoria Stilwell Academy Dog Trainer Course (VSA-DTC)	12 (1.69%)

Note. Respondents were shown list of training programs if they selected the options “Yes, from graduating from a training program” (209, 29.60%) or “Yes, from both a training program and a credentialing board” (191, 27.05%). The total number of programs was calculated as 138 by adding the 17 programs proposed by the research team plus 121 added by respondents as write-in options. ^a^ These credentials were mentioned as write-in options by >1% of the sample.

**Table 2 animals-15-01255-t002:** Credentialing board certification identified by 1% or more of respondents as a follow-up to the question “Have you obtained any dog training certifications?”.

Organization	Name of Credential	Number of Respondents
Certification Council for Professional Dog Trainers	Certified Professional Dog Trainer-Knowledge Assessed (CPDT-KA)	162 (22.94%)
Certification Council for Professional Dog Trainers	Certified Professional Dog Trainer-Knowledge and Skills Assessed (CPDT-KSA)	12 (1.69%)
International Association of Animal Behavior Consultants	Certified Behavior Consultant Canine-Knowledge Assessed (CBCC-KA) ^a^	13 (1.84%)
International Association of Canine Professionals	Certified Dog Trainer (IACP-CDT)	17 (2.40%)
International Association of Animal Behavior Consultants	Certified Dog Behavior Consultant (IAABC-CDBC)	29 (4.10%)
International Association of Animal Behavior Consultants	Accredited Dog Trainer (IAABC-ADT) ^a^	12 (1.69%)
Pet Professional Guild	Professional Canine Behavior Consultant (PCBC-A)	9 (1.27%)
Pet Professional Guild	Professional Canine Trainer (PCT-A)	13 (1.84%)

Note. Respondents were shown these options if they selected the options “Yes, from taking an exam or evaluation from a credentialing board” (112, 15.86%) or “yes, from both a training program and a credentialing board” (191, 27.05%). The total number of credentials was calculated as 39 by adding the 9 proposed by the research team plus 31 added by respondents as write-in options. ^a^ These options included “International Association of Animal Behavior Consultants (IAABC): Certified Behavior Consultant Canine-Knowledge Assessed (CBCC-KA)”, which is incorrect. There were 12 instances in which respondents wrote in the correct IAABC certificate “International Association of Animal Behavior Consultants: Accredited Dog Trainer (ADT)”, so this option was added to the list.

**Table 3 animals-15-01255-t003:** Summary of moderate and strong associations with opinion on regulation.

	Trainers More Likely to	Trainers Less Likely to
Support regulation	Have completed training programs (V = 0.2) Support regulations being led by welfare organizations (V = 0.26) Use the term “force-free” (V = 0.26) Use the term “science-based” (V = 0.21)	Use the term “balanced” (V = 0.27) Use electronic (V = 0.3) and prong (V = 0.31) collars
Undecided about regulation	Use the term “balanced” (V = 0.29) Use choke (V = 0.24), electronic (V = 0.42) and prong (V = 0.33) collars	Have completed training programs (V = 0.23) Support regulations being led by welfare organizations (V = 0.32) Support regulations being led by the government (V = 0.32) Use the term “force-free” (V = 0.3) Use the term “science-based” (V = 0.3)
Against regulation	Use the term “balanced” (V = 0.63) Use the term “obedience” (V = 0.46) Use the term “pack leader” (V = 0.47) Use choke (V = 0.44), electronic (V = 0.74) and prong (V = 0.77) collars	Completed training programs (V = 0.44) Hold dog training certifications (V = 0.34) Support regulations being led by welfare organizations (V = 0.53) Support regulations being led by the government (V = 0.32) Use the term “force-free” (V = 0.67) Use the term “science-based” (V = 0.44)

Note. Cramer’s V interpretation [30]: negligible (<0.10), weak (0.10–0.20), moderate (0.20–0.40), relatively strong (0.40–0.60), strong (0.60–0.80), and very strong (>0.80).

**Table 4 animals-15-01255-t004:** Summary of moderate and strong associations with recommending veterinary consultation.

	Trainers More Likely to	Trainers Less Likely to
Recommend veterinary consultation	Live in Alberta (V = 0.34) Be in favor of regulation (V = 0.23)	
Not recommend veterinary consultation	Live in Ontario (V = 0.25) Be against regulation (V = 0.7) Use the term “balanced” (V = 0.6) Use the term “obedience” (V = 0.43) Use the term “pack leader” (V = 0.54)	Use the term “science-based” (V = 0.4) Use the term “force-free” (V = 0.53)

Note. Cramer’s V interpretation [30]: negligible (<0.10), weak (0.10–0.20), moderate (0.20–0.40), relatively strong (0.40–0.60), strong (0.60–0.80), and very strong (>0.80).

## Data Availability

The original data presented in the study are openly available via OSF: [https://osf.io/s5qmz/?view_only=97f2a84b5c5242f9875a85ca7bff9b7e] (accessed on 28 March 2025).

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
