# Peer review of "A Survey of the Professional Characteristics and Views of Dog Trainers in Canada"

_animals, 2025, doi:10.3390/ani15091255_

Round 1
Reviewer 1 Report
Comments and Suggestions for Authors
Overall, I thought the manuscript was well-written. The lack of regulation for dog trainers is becoming an increasingly important issue as researchers recognise the potential impacts that dog training may have on human-animal interactions and the welfare of companion animals. While the study is based in Canada, its findings are highly relevant to companion dogs worldwide. Therefore, it is a useful contribution to the literature as the manuscript adds to our understanding of dog trainers’ characteristics and views. I would recommend acceptance with a few minor edits.
Introduction
Line 69-70: “Moreover, there is disagreement on which methods should be regarded as humane [18], and whether the use of aversive-based training is justified in certain cases.” Please can you expand here, so the reader does not have to refer to the reference to know what point is being raised here.
Methods
Lines 152-153: Was anything learnt during the piloting?
Results
Lines 224-227: “We compared these results with population estimates in Canada [32] to understand 224 the general representativeness of sample. Our sample may be overrepresented for B.C. 225 and underrepresented for Quebec and Ontario. However, this comparison is approximate 226 as general population estimates may differ from the actual distribution of dog trainers 227 across provinces.” I don’t suppose there are dog populations estimates that you could also include here? I am guessing probably not?
Lines 280-338: The credentials mentioned <1% - I feel that these could be listed in supplementary materials and do not need to be in the paper itself unless the authors/editors feel very strongly about this. (Same with lines 350-366.
Line 275: “Credentials mentioned 6 times: Leerburg Academy (Michael Ellis courses).” Why is this line by itself please? A paragraph with just one sentence or is some text missing please?
Tables 1 and 2 seems to be missing their bottom lines?
Out of interest, what was the rational behind the behaviours that were grouped as “poor manners”? Could you add a sentence about this into the paper (probably in the Methods section)?
I would appreciate it if you could add in a definition of what you meant by reactivity to dogs/people? Where the trainers provided a definition or was this left for the trainer to interpret? (This could be added in to save the reader having to stop reading to look at the survey.)
Just a comment, as I am based in the UK, it is fascinating to read the ‘Training method descriptions’ section as lots of terms are used in Canada which are not used in the UK. It’s great that you thought to include this.
I noted rather a lot of chi-squared tests where run on the same dataset. Did you consider correcting for the multiple comparisons but using something like Bonferroni correction please?
Discussion
In general, the Discussion was well thought out. However, unfortunately, I found there was a rather a lot of repetition of the Results section - for example, lines 778-785, 837-844. 894-898 are all unnecessary in my opinion. As the Discussion is quite long, I would highly recommend going through and removing the lines that are replications of the Results section. The read can refer back to the tables/text if needed.
Author Response
Thank you for your comments! Please find the detailed responses below and the corresponding revisions highlighted in the re-submitted files.
Introduction
Line 69-70: “Moreover, there is disagreement on which methods should be regarded as humane [18], and whether the use of aversive-based training is justified in certain cases.” Please can you expand here, so the reader does not have to refer to the reference to know what point is being raised here.
More information has been added to L 69-74.
Methods
Lines 152-153: Was anything learnt during the piloting?
Feedback from trainers helped us with some of the wording so we could be better aligned to industry-standard language (we added a note on this in L 161-165).
This feedback also led us to edit some questions for clarity and include items on the list of tools in question 20. In the case of the French version of the survey, we received feedback on the quality/clarity of the translation from bilingual trainers.
Results
Lines 224-227: “We compared these results with population estimates in Canada [32] to understand 224 the general representativeness of sample. Our sample may be overrepresented for B.C. 225 and underrepresented for Quebec and Ontario. However, this comparison is approximate 226 as general population estimates may differ from the actual distribution of dog trainers 227 across provinces.” I don’t suppose there are dog populations estimates that you could also include here? I am guessing probably not?
There is data indicating there were 7.9 million dogs in Canada in 2022 (Canadian Animal Health Institute), but we felt it would not add to this discussion as this is not disaggregated across provinces.
Lines 280-338: The credentials mentioned <1% - I feel that these could be listed in supplementary materials and do not need to be in the paper itself unless the authors/editors feel very strongly about this. (Same with lines 350-366).
Thank you for your suggestion. These have been moved to Supplementary Material.
Line 275: “Credentials mentioned 6 times: Leerburg Academy (Michael Ellis courses).” Why is this line by itself please? A paragraph with just one sentence or is some text missing please?
Tables 1 and 2 seems to be missing their bottom lines?
We apologise for this; it seems there was some sort of formatting issue and the notes of the tables became misaligned. In any case, the write-in options have been moved to Supplementary Material as per the previous suggestion.
Out of interest, what was the rational behind the behaviours that were grouped as “poor manners”? Could you add a sentence about this into the paper (probably in the Methods section)?
I would appreciate it if you could add in a definition of what you meant by reactivity to dogs/people? Where the trainers provided a definition or was this left for the trainer to interpret? (This could be added in to save the reader having to stop reading to look at the survey.)
This was based on pilot feedback (added as mentioned above in L. 161-165). We wanted to provide terminology aligned with industry standards, where “manner” classes are popular, and the term “reactivity” is increasingly more common.
Stephens-Lewis et al. (2024) recently indicated that the label “reactivity” is becoming more popular across lay and professional contexts to refer to dogs broadly presenting with problematic behaviours. They highlighted the lack of consensus about what is exactly encompassed in this term, even considering it as a “catch-all” for various behaviours.
We were intentionally a bit vague to leave room for flexibility in the interpretation of each trainer. While we acknowledge this can create some difficulty generalizing the data, we considered it was a better approach than providing rigid definitions that may not align with individual trainer’s perspectives.
Just a comment, as I am based in the UK, it is fascinating to read the ‘Training method descriptions’ section as lots of terms are used in Canada which are not used in the UK. It’s great that you thought to include this.
Thank you!
I noted rather a lot of chi-squared tests where run on the same dataset. Did you consider correcting for the multiple comparisons but using something like Bonferroni correction please?
Discussion
In general, the Discussion was well thought out. However, unfortunately, I found there was a rather a lot of repetition of the Results section - for example, lines 778-785, 837-844. 894-898 are all unnecessary in my opinion. As the Discussion is quite long, I would highly recommend going through and removing the lines that are replications of the Results section. The read can refer back to the tables/text if needed.
Thank you for your suggestion. These lines (and others), were removed from the Discussion to reduce repetitiveness and facilitate reading.
Reviewer 2 Report
Comments and Suggestions for Authors
This is an important manuscript that helps clarify and summarize the overall status of the professional dog training situation in Canada, including overall qualifications, training offerings, training methods, education, background, and more. This study addressed the characteristics and methods of dog trainers across the country, along with their beliefs and descriptions about their own programs and thoughts on the state of the profession. Overall, this manuscript did an excellent job highlighting the current dog training situation, and provides an important picture of the current landscape of dog training, which will help provide context on which to base the development of new programs and training to encourage optimal training guidelines to the benefit of trainers, dog owners, and the dogs themselves.
I have including the following comments and suggestions to help strengthen the manuscript:
Lines 61-66: This was a great paragraph to include here, laying the background for why this type of training is more favorably viewed.
Lines 103-104: Are there any countries or locations to your awareness where the training profession is better standardized or regulated? If so, it may be good to include as an example here in this discussion.
Lines 116-118: Is there any postulation as to why animal trainers are the most averse to accreditation? Is it associated with their higher likelihood to use aversive training methods?
Lines 250-251: Were there any details about what “self-educated” may have consisted of (e.g., reading online vs shadowing a fellow trainer or other direct exposure to the field)?
Lines 257-268: Are there any details about the methods taught by most of these training programs? Do most of them teach exclusively positive-reinforcement based training method, vs. balanced or aversive? This discussion would also fit in well in Lines 786-799.
Lines 375-389: Did you look to see if there was any association between location of training (e.g., settings where the clients are present, vs. settings where trainer is with the dog/dogs absent the client) and primary training methods (e.g., including more aversive techniques)
Lines 466 – 474: It would be interesting to know the reasoning behind the use of typically considered “aversive” items by the trainers that use them – do they think they are essential despite being aversive for certain dogs, or do some have differing opinions about whether some of these items are actually “aversive” and therefore are more inclined to use them.
Lines 876-882: This is a very interesting, and also important finding. Is there any hypothesis as to why “balanced trainers” would be less likely to think medication would ever be needed compared to the views of other trainers?
Lines 887-889: This is a very important point to emphasize here, as it really highlights the influence and impact trainers may have on overall animal health, and the esteem that their clients view them with. This really helps highlight why the characterization provided by this manuscript is so relevant.
Lines 938 – 940: Is there any speculation as to the association/directionality between certification and more reward-based methods? Is there any indication if education and certification processes teach trainers to use more reward-based methods, or are those who want to use such methods more likely to seek out training and certification?
Author Response
Thank you for your comments! Please find the detailed responses below and the corresponding revisions highlighted in the re-submitted files.
Lines 61-66: This was a great paragraph to include here, laying the background for why this type of training is more favorably viewed.
Thank you!
Lines 103-104: Are there any countries or locations to your awareness where the training profession is better standardized or regulated? If so, it may be good to include as an example here in this discussion.
We are not aware of any countries which currently implement such regulations. The use of aversive-based collars is prohibited in some countries, but we chose not to include this information as it substantially increased the manuscript length and citation load without meaningfully contributing to the central focus of our study.
Lines 116-118: Is there any postulation as to why animal trainers are the most averse to accreditation? Is it associated with their higher likelihood to use aversive training methods?
The authors did not provide potential explanations for this finding. As mentioned in L 847-858 in relationship to our current data, more research is needed to explore whether the likelihood of supporting professional regulation is influenced more by the methods employed by trainers or by their education and certification (and the directionality of these relationships).
Lines 250-251: Were there any details about what “self-educated” may have consisted of (e.g., reading online vs shadowing a fellow trainer or other direct exposure to the field)?
This has been added L 258-259.
Lines 257-268: Are there any details about the methods taught by most of these training programs? Do most of them teach exclusively positive-reinforcement based training method, vs. balanced or aversive? This discussion would also fit in well in Lines 786-799.
As mentioned in L 765-769- most of the programs are from organizations promoting reward-based training or LIMA approaches. However, we also wanted to underscore the great abundance and variability of programs and credentials and the difficulty of categorizing their training method. Many programs do not provide any easily available information that would allow us to assess their underlying methodology, and together with the ongoing disagreement regarding what is considered reward-based or humane training, our assessments may be viewed as subjective or biased. This illustrates our point on how confused guardians may get when selecting trainers.
Lines 375-389: Did you look to see if there was any association between location of training (e.g., settings where the clients are present, vs. settings where trainer is with the dog/dogs absent the client) and primary training methods (e.g., including more aversive techniques)
This is a great suggestion, thank you! Unfortunately, we did not have space to explore all possible associations in our data, and we wanted to limit the number of comparisons to reduce the risk of errors associated with multiple testing.
Lines 466 – 474: It would be interesting to know the reasoning behind the use of typically considered “aversive” items by the trainers that use them – do they think they are essential despite being aversive for certain dogs, or do some have differing opinions about whether some of these items are actually “aversive” and therefore are more inclined to use them.
This is also a great idea for future research.
Lines 876-882: This is a very interesting, and also important finding. Is there any hypothesis as to why “balanced trainers” would be less likely to think medication would ever be needed compared to the views of other trainers?
More information was added in L 796-800. The authors also found that balanced trainers considered separation anxiety to be preventable and caused by human behaviour. They suggested this could lead them to consider it could be treated by modifying human behaviour and thus not needing medication.
Lines 887-889: This is a very important point to emphasize here, as it really highlights the influence and impact trainers may have on overall animal health, and the esteem that their clients view them with. This really helps highlight why the characterization provided by this manuscript is so relevant.
Thank you!
Lines 938 – 940: Is there any speculation as to the association/directionality between certification and more reward-based methods? Is there any indication if education and certification processes teach trainers to use more reward-based methods, or are those who want to use such methods more likely to seek out training and certification?
As mentioned in L 847-858 more research is needed to explore this issue. We prefer to avoid speculation beyond the scope of our findings.
Reviewer 3 Report
Comments and Suggestions for Authors
The research aimed to evaluate the way of dog trainers in Canada, through an online questionnaire.
The topic is quite current because in many countries the same situation is found regarding the training of dog trainers that is not formalized.
The introduction is complete and offers a clear picture of the situation of dog training in Canada.
The section of materials and methods explains in detail the phases that led to the final version of the questionnaire.
The results are expressed in excessive detail and this makes reading this section quite difficult: for example the list of credentials could be eliminated and possibly inserted in the supplementary material. The results as a whole are interesting and clarify many aspects of the profession of dog trainer.
The discussion examines the results in detail and indicates the critical points of the research. The conclusion is in line with what is expressed in the results.
the bibliography is extensive and updated.
In my opinion the paper can be accepted by requesting the authors to make it easier to read, eliminating some non-essential data
Author Response
The research aimed to evaluate the way of dog trainers in Canada, through an online questionnaire.
The topic is quite current because in many countries the same situation is found regarding the training of dog trainers that is not formalized.
The introduction is complete and offers a clear picture of the situation of dog training in Canada.
The section of materials and methods explains in detail the phases that led to the final version of the questionnaire.
The results are expressed in excessive detail and this makes reading this section quite difficult: for example the list of credentials could be eliminated and possibly inserted in the supplementary material. The results as a whole are interesting and clarify many aspects of the profession of dog trainer.
The discussion examines the results in detail and indicates the critical points of the research. The conclusion is in line with what is expressed in the results.
the bibliography is extensive and updated.
In my opinion the paper can be accepted by requesting the authors to make it easier to read, eliminating some non-essential data
Thank you for your comments. The results section has been revised for clarity and the lists of write-in credentials have been moved to supplementary material to facilitate reading.
Reviewer 4 Report
Comments and Suggestions for Authors
The manuscript “A survey of the professional characteristics and views of dog trainers in Canada” addresses a significant issue in the dog training industry, highlighting the lack of regulation and standardization for dog trainers. This is a problem not only in Canada but in many other countries. The topic is important because trainers have a significant impact on the behavior and well-being of dogs and have a huge impact on how handlers interact with their dogs. As the study focuses on an under-regulated field, its findings are particularly relevant to animal welfare organisations, policy makers and dog owners. The discussion of the diversity of training qualifications and methods provides up-to-date insight into this market. The survey had a large number of respondents (706 valid responses) and collected data on various aspects, including demographics, qualifications and training methods. The wide range of data suggests that the study offers a comprehensive overview of the current state.
My comments and suggestions:
- Although the Introduction outlines the differences between reward-based and aversion-based methods, a more precise definition of these terms at an early stage would have helped to provide clarity.For example, explicitly stating the operational criteria for "balanced training" versus "aversion-based techniques" could have avoided ambiguity later when discussing the survey results.
- It would be helpful to add a brief explanation of how the potential adoption of regulatory practices may impact long-term outcomes for dogs, handlers, and trainers.
- The article highlights that the diversity of qualifications and methods can pose a challenge for dog owners in selecting competent trainers.Expanding on this point with potential strategies or frameworks for assessing trainer qualifications could increase the practical utility of the study.
- Consideration could be given to dividing the discussion into subsections/sections, which would improve readability and make it easier for different audiences (scientists, practitioners, decision-makers) to find relevant information.
The manuscript meets the requirements for publication in the journal Animals and can be published after minor revision.
Author Response
Thank you for your comments! Please find the detailed responses below and the corresponding revisions highlighted in the re-submitted files.
Although the Introduction outlines the differences between reward-based and aversion-based methods, a more precise definition of these terms at an early stage would have helped to provide clarity. For example, explicitly stating the operational criteria for "balanced training" versus "aversion-based techniques" could have avoided ambiguity later when discussing the survey results.
We intentionally chose terms that were popular in the dog training industry, recognizing they may leave some room for ambiguity. As noted throughout the manuscript, there is ongoing debate on terminology, and different trainers could interpret these differently. Given that our approach relied on self-identification, participants were able to apply their own understanding to these terms when indicating how likely they would be to use them to describe their training methods.
Nevertheless, in the case of balanced training we do mention it is generally considered as a blend of using aversive-based techniques and rewards (L 58-60: “Many trainers who incorporate aversive-based techniques also use rewards, and this blend of methods is colloquially referred to as “balanced training”).
It would be helpful to add a brief explanation of how the potential adoption of regulatory practices may impact long-term outcomes for dogs, handlers, and trainers.
This was added in L 872-875.
The article highlights that the diversity of qualifications and methods can pose a challenge for dog owners in selecting competent trainers. Expanding on this point with potential strategies or frameworks for assessing trainer qualifications could increase the practical utility of the study.
This was clarified in L 727-730.
Consideration could be given to dividing the discussion into subsections/sections, which would improve readability and make it easier for different audiences (scientists, practitioners, decision-makers) to find relevant information.
Thank you for your suggestion. We considered it but we chose not to divide the discussion, as we believe the issues are interconnected and relevant to all audiences. Separating them might obscure some important overlaps between perspectives. However, we did rewrite parts of the discussion to make it more straightforward, reducing repetitiveness and increasing readability.